# Amino Acid Substitution within Seven-Octapeptide Repeat Insertions in the Prion Protein Gene Associated with Short-Term Course

**DOI:** 10.3390/v14102245

**Published:** 2022-10-13

**Authors:** Zhongyun Chen, Haitian Nan, Yu Kong, Min Chu, Li Liu, Jing Zhang, Lin Wang, Liyong Wu

**Affiliations:** Department of Neurology, Xuanwu Hospital, Capital Medical University, Beijing 100053, China

**Keywords:** Creutzfeldt–Jakob disease, genetics, prion disease, octapeptide repeat insertion, amino acid substitution

## Abstract

The majority of seven-octapeptide repeat insertion (7-OPRI) carriers exhibit relatively early onset and a slowly progressive course. We have presented three cases of 7-OPRI, including two that are rapidly progressing, and compared the clinical and ancillary characteristics of the short-term and long-term disease course, as well as factors that influence disease course. The clinical and ancillary features of three new 7-OPRI patients in a Chinese pedigree were analyzed. Global data on 7-OPRI cases were then collected by reviewing the literature, and the cases were grouped according to clinical duration as per the WHO sCJD criteria, with a two-year cut-off. A Chinese pedigree has a glycine-to-glutamate substitution within the 7-OPRI insertion, which enhances the hydrophilicity of the prion protein. Two cases in this pedigree had a short disease course (consistent with the typical clinical and ancillary features of sCJD). In addition, the members of this pedigree had a later onset (*p* < 0.001) and shorter disease course (*p* < 0.001) compared to previously reported 7-OPRI cases with 129 cis-M and a similar age of onset and disease course to that of cases with 129 cis-V. The 7-OPRI cases with a shorter clinical course (*n* = 4) had a later onset (*p* = 0.021), higher rate of hyperintensity on MRI (*p* = 0.029) and higher frequency of 129 cis-V (*p* = 0.066) compared to those with a longer clinical course (*n* = 13). The clinical presentation of 7-OPRI is significantly heterogeneous. Codon 129 cis-V and amino acid substitution within repeat insertions are possible contributors to the short-term disease course of 7-OPRI.

## 1. Introduction

Genetic prion diseases (gPrDs), which are caused by pathological sequence variations in the prion protein gene (*PRNP*), account for 10–15% of all prion diseases [1]. The majority of *PRNP* mutations are point mutations, although there have been reports of octapeptide repeat insertions (OPRIs) and octapeptide repeat deletions (OPRDs) in the *PRNP* octapeptide repeat region, which is located between codons 51 and 91.

GPrDs have been reported to be caused by OPRIs with 2 to 12 repeats. While OPRIs do not directly affect prion protein (PrP) conformation, functional studies have shown that normal prion protein (PrP^C^) with OPRIs form protease-resistant PrP more rapidly and aggregate more readily, thereby promoting the formation of a pathological isoform known as PrP^Sc^ [2,3]. In addition, the octapeptide region of *PRNP* binds to copper ions, and OPRIs can alter copper binding, which may impair the copper-mediated functions of PrP^C^ or reduce copper-dependent resistance to PrP^Sc^ conversion [4]. Patients with OPRIs show significant clinical and neuropathological heterogeneity, depending on the number of repeats. OPRIs with 2 to 4 repeats tend to resemble sporadic Creutzfeldt–Jakob-disease (sCJD) clinically with a later age of onset and shorter clinical duration, and those with 5 to 7 repeats are more frequently associated with CJD and occasionally with Gerstmann–Straussler–Scheinker syndrome (GSS), albeit with earlier onset and longer disease duration compared to typical sCJD. Cases with 8-OPRI to 12-OPRI are more frequently associated with the GSS phenotype [1,3,5]. However, this phenotypic classification based on the number of OPRI repeats is not always accurate [6]. Genetic polymorphisms at *PRNP* codon 129 and pathological changes also contribute to the disease phenotype [1,3,5,7].

Only 6 families and 14 patients with 7-OPRI have been reported to date, and although the phenotype of these mutation carriers varies significantly across and within families, the majority present with relatively early onset and a slowly progressive course of the disease [7,8,9,10,11,12,13,14,15,16,17]. Generally, the insertion sequence in the octapeptide repeat region is fairly conserved, and the base changes do not affect the amino acid sequence. Here, we present three new 7-OPRI patients from the same family first reported in 2007 who had an amino acid substitution (AAS) within OPRIs [15,16]. These new cases differ from previously reported cases that were characterized by relatively late onset and rapid progression. We also reviewed the published literature on 7-OPRI cases to further explore disease heterogeneity and classified them on the basis of clinical duration longer or less than 2 years as per the WHO sCJD criteria. The clinical and ancillary features of short and long-term disease courses were compared, and the factors influencing the course in 7-OPRI patients were screened.

## 2. Methods

### 2.1. Study Design

The family members with 7-OPRI were enrolled at the Department of Neurology of Xuanwu Hospital. The PubMed, Embase and Web of Science databases were searched on May 2022 for primary research articles and case studies reporting 7-OPRI carriers.

### 2.2. Clinical and Laboratory Data

The following demographic and clinical variables were extracted from the retrieved articles: age at onset (years), gender, duration of symptoms (years), family history, initial symptoms, neurological manifestations during the clinical course, and auxiliary examination results, including periodic sharp wave complexes (PSWCs) on electroencephalogram (EEG), cerebrospinal fluid (CSF) 14-3-3 and tau protein, real-time quaking-induced conversion (RT-QuIC), codon 129 polymorphism, neuroimaging, and neuropathological findings. Patients were further grouped on the basis of the course of disease (>2 years and ≤2 years). Cis polymorphism of codon 129 is located on the same allele as the mutation and has a greater impact on disease manifestation, while the trans polymorphism has a smaller (or possibly no) impact [18].

### 2.3. Laboratory Methods

#### 2.3.1. Genetic Analyses

Genomic DNA was extracted from fresh peripheral blood leukocytes, and whole-exome sequencing (WES) libraries were generated using the Agilent SureSelect Human All Exon V6 Kit (Agilent Technologies, Santa Clara, CA, USA). The detailed procedure has been described in our previous study [19].

*PRNP* open reading frame was amplified from the genomic DNA by polymerase chain reaction (PCR). The primer sequences were as follows: forward 5′-CCATTGCTATGCACTCATTCA-3′, reverse 5′-AGAAAGAGTGAGACACCACCA-3′. Nested sequencing primers were used to sequence a small exonic PCR amplicon that included the *PRNP* octapeptide repeat region; forward 5′-GACCTGGGCCTCTGCAAGAAGCGC-3′, reverse 5′-GGCACTTCCCAGCATGTAGCCG-3′. The PCR-amplified fragments were verified by 2% agarose electrophoresis, cleaved with Not I and Sal I restriction endonucleases and ligated into the plasmid as previously described [8]. Colonies containing the recombinant plasmids were selected and sequenced in both directions. The methionine/valine polymorphism at codon 129 (rs1799990) was determined from sequence data.

#### 2.3.2. CSF 14-3-3 Protein Level Test and RT-QuIC Assay

CSF 14-3-3 protein level measurement and the RT-QuIC assay were performed at the CDC National Reference Laboratory for Human Prion Diseases in China as previously described [20,21].

#### 2.3.3. CSF Tau Level Test

Total tau protein level in CSF samples was measured by enzyme-linked immunosorbent assay (ELISA) (Innotest Htau Ag; Fujirebio, Belgium), and a Tau level above 1400 pg/mL was considered positive.

#### 2.3.4. Electroencephalogram

The CJD subjects received 2-h EEG (Micromed, Treviso, Italy), and electrodes were placed according to the international 10–20 system. PSWCs were defined according to the criteria for PSWCs published in 1996 [22].

#### 2.3.5. Magnetic Resonance Imaging

All MRI scans were performed at 3.0 T (Erlangen, Germany). Hyperintensity was assessed using diffusion-weighted imaging (DWI) and T2 fluid-attenuated inversion recovery in the cortex or basal ganglia.

#### 2.3.6. Positron Emission Tomography

PET scans were performed using a GE Signa PET/MR 3.0 Tesla scanner (GE Healthcare, Milwaukee, WI, USA). The ^18^F-FDG-PET images were acquired within 15 min after intravenous injection of ^18^F-FDG (~308 MBq) with an uptake time of 30 min.

#### 2.3.7. Statistical Analysis

Statistical analyses were performed using SPSS version 22.0 (IBM, Armonk, NY, USA). Continuous data are represented as the mean ± SD and were compared using *t*-tests. Dichotomous data are presented in the form of percentages and were compared using Fisher’s test. A two-tailed *p*-value ≤ 0.05 was considered statistically significant. Tukey’s multiple comparisons tests were performed to compare the data among different neuropathological change groups using GraphPad Prism version 7 software.

## 3. Results

### 3.1. Clinical Presentation of Each Patient

Six subjects from the examined family presented with dementia (Figure 1). The clinical details of patients III-4, III-7 and IV-1 have been reported. Genetic analysis was performed for patients III-2, III-4, III-7 and IV-1.

### 3.2. Subject IV-1

The patient was hospitalized in their 40s for exhibiting dementia over a week. Neurological examination indicated cognitive deficits based on a score of 19 on the Mini-mental State Examination and 10 on the Montreal Cognitive Assessment. Other neurological tests were negative. The brain MRI DWI suggested bilateral symmetrical hyperintensity in the fronto-temporoparietal-occipital cortex and subcortex (Figure 2A,B). The ^18^F-fluorodeoxyglucose positron emission tomography (FDG-PET) also showed significant hypometabolism in the aforementioned areas (Figure 2C,D). The EEG revealed no specific changes and no PSWCs. CSF 14-3-3 protein and RT-QuIC were positive, and the CSF tau protein was significantly elevated. The patient’s condition rapidly deteriorated in the months following the hospital visit, and the patient exhibited symptoms of extrapyramidal and pyramidal damage, as well as severe psychiatric and behavioral abnormalities. Six months after the onset of the disease, the patient progressed to akinetic mutism.

### 3.3. Subject III-4

The patient presented with executive dysfunction in their 50s. A neurological examination two weeks after the onset of the disease revealed impaired memory, orientation, calculation and execution, while other neurological tests were negative. Brain MRI DWI showed bilateral frontotemporal parietal occipital hyperintensity, with the right side being more prominent (Figure 2E,F). An EEG showed diffuse slow waves without PSWCs. CSF 14-3-3 protein and RT-QuIC were positive, and CSF tau protein levels were significantly elevated. In the following months, the patient’s symptoms progressively worsened, and the patient presented with ataxia and speech impairment. The patient became increasingly bedridden in the last 4 months and died 18 months after the onset of symptoms.

### 3.4. Subject III-7

This patient presented with forgetfulness in their 60s. Four years after the onset of symptoms, the patient’s cognitive status had deteriorated significantly, manifesting as relative ignorance, a decline in the ability to perform daily activities, and severe mental and behavioral abnormalities. Brain MRI showed brain atrophy without hyperintensity (Figure 2G,H). A heterozygous mutation (c.1752 + 3G > A) was found in the Alzheimer’s disease (AD) susceptibility gene NOS3, and AD was suspected. The patient presented with ataxia, tremors, and occasional speech impairment in the last 6 months of his life. The patient had gross nystagmus when looking to the right, increased limb tension, hyperreflexia, and positive pathological signs on neurological examination. The patient became increasingly bedridden over the last two months and died 5.5 years after the onset of symptoms.

### 3.5. Subject III-2

The patient is the mother of the proband, and her specific clinical information has been reported previously [15,16].

### 3.6. Subjects I-1 and II-2

I-1 experienced dementia and paralysis in their 50s and passed away around 6 months later. II-2 experienced involuntary movements and progressive dementia in their 50s. The patient was clinically diagnosed with suspected cerebellar atrophy and died six months after the onset of the disease.

### 3.7. Genetic Analysis

Patients IV-1, III-2, III-7 and III-4 were homozygous for methionine (M) at *PRNP* codon 129. Sequencing of this region demonstrated a 168-bp insertion in the octapeptide repeat region [R1-R2-R2-R2-R2-R2-R3g-R2-R3g-PHGGGWEQ-R3-R4] (Figure 3A). The last repeat of *PRNP* insertion had a G > A substitution at the seventh triplet (from GGG to GAG), leading to conversion from glycine (hydrophobic residue) to glutamic acid (acidic residue). A 3D structure model of the octapeptide repeat based on the nuclear magnetic resonance structure of the human prion protein at pH 6.2 (Protein Data Bank accession code: 1OEH) is shown in Figure 3B,C [23]. The proposed structure includes the octapeptide repeats (protein 61–68). Therefore, the highly conserved amino acid sequence PHGGGWGQ of R2 and the mutated form of PHGGGWEQ are structured.

### 3.8. Analysis of Combined Data

A total of 14 patients with 7-OPRI were identified from the literature. The clinical and auxiliary characteristics of the three new and the previously reported 7-OPRI patients are summarized in Appendix A. There was significant phenotypic heterogeneity among the patients. The age at onset ranged from 18 to 60 years (mean 40.6 years, SD = 14.1 years). The disease lasted from 0.6 to 16 years (mean 7.3 years; SD = 5.2 years). The most common initial symptoms were cognitive deficits, psychiatric disturbances and ataxia. Cognitive dysfunction (100%, 17/17), psychiatric disturbances (76.5%, 13/17) and extrapyramidal symptoms (70.6%, 12/17) were the most common symptoms during the course of the disease. Hyperintensity on MRI and PSWCs on EEG were 41.9% (3/7) and 18.2% (2/11) respectively. Fourteen patients were genotyped for codon 129, of which 12 were identified as cis-M (6 with 129 MM, 5 with 129 cis-M (trans unknown) and 1 with 129 MV) and 2 as cis-V (1 with 129 VV and 1 with 129 MV). Autopsies were performed on 11 patients. Nine patients had mild to severe spongiosis, three had multicentric plaques in the cerebellum and cerebral cortex, three had elongated plaques in the neocortex or cerebellum, and one did not exhibit any specific histopathological changes.

### 3.9. Comparison of Clinical and Ancillary Features between Cases with Short-Term and Long-Term Course

The clinical and ancillary features of the long-term and short-term disease courses are summarized in Table 1. Patients with a short clinical course had a later onset (53 ± 7.3 vs. 35 ± 12.9 years, *p* = 0.021) and a higher rate of hyperintensity on MRI (3/3 vs. 0/4, *p* = 0.029) compared to those with a long clinical course. Patients with shorter clinical courses also tended to have higher rates of 129 cis V, albeit without statistical significance (*p* = 0.066). The majority of patients with long-term illnesses will eventually develop the clinical characteristics of prion diseases, even though they are absent in the early stages of the disease.

### 3.10. The Effect of Codon 129 Genotype and AAS within OPRIs on Disease Course

Patients were grouped according to the codon 129 genotype and AAS within OPRIs (Figure 4). Cases in our pedigree (129 cis-M with AAS within OPRIs), including the case published in 2007, had a later onset (mean difference 24.5 years, *p* = 0.0001) and shorter disease duration (mean difference 9.0 years, *p* = 0.0004) compared to previously reported 7-OPRI cases with 129 cis-M without AAS within OPRIs, and a similar age of onset and disease course to that of patients with 129 cis-V. These differences persisted when compared with 129 cis-M with AAS within OPRIs and 129 cis-V cases. However, there was no difference in the duration of symptoms between 129 cis-M and 129 cis-V cases, while the 129 cis-M cases tended to have an earlier age of onset (*p* = 0.0707).

### 3.11. The Effect of the Neuropathological Changes on the Course of the Disease

There was no difference in the length of symptom duration among the different pathological subgroups. However, patients with co-existing spongiosis and multicentric plaques had a later age of onset compared to those with spongiosis alone (*p* = 0.0365) and those with spongiosis and elongated plaques (*p* = 0.0048) (Appendix A). Nevertheless, given the small sample size, these findings need to be carefully interpreted.

## 4. Discussion

In this report, we described three patients from a Chinese family with an AAS within 7-OPRI in the *PRNP* gene. Patients in this pedigree are clinically heterogeneous, with a later age of onset and a shorter duration of disease compared to previously reported cases. Our study shows that 7-OPRI patients with a shorter clinical course have a later onset, higher rate of hyperintensity on MRI and higher frequency of 129 cis V. The scant evidence presented herein suggests that codon 129 cis-V and AAS within OPRIs are possible contributors to the short-term disease course.

Although carriers of the 7-OPRI mutation exhibit a wide range of phenotypic characteristics, most have relatively early-onset and slow progression. Only 2 of the 14 published cases of 7-OPRI showed rapid progression, and both cases belonged to the same family [7]. GSS was pathologically diagnosed in one of these patients with ataxia as the predominant phenotype. In our pedigree, two patients presented with predominantly rapidly progressive dementia with typical MRI features of CJD in the early stage, and the mother of the proband was pathologically diagnosed with CJD, which differed from those previously described in the literature. The patients with a shorter disease course were older and more closely resembled the clinical phenotype of CJD.

In the years preceding acute deterioration, most patients with 7-OPRI only exhibit slowly progressive cognitive impairment and psycho-behavioral abnormalities, with no specific ancillary tests to support prion diseases. As a result, these patients are frequently misdiagnosed in the absence of a clear family history of CJD [8,11]. Subject III-3 was a representative case in our pedigree. For years before the exacerbation of symptoms, this patient had mild cognitive impairment and mild to moderate psycho-behavioral abnormalities. Given the family history of dementia (not known to be CJD), the patient was screened for genes linked to AD and frontotemporal dementia, but no causative genes were found. This also led to the patient being misdiagnosed with AD for years until the proband mutation was detected. Therefore, it is recommended to screen for *PRNP* mutations in patients with genetically undefined dementia or psychiatric symptoms.

Codon 129 is a determinant of the disease susceptibility and phenotype of PrDs [24,25]. However, the effect of codon 129 polymorphism on the age of onset and duration of disease varies among PrDs. For example, V carriers with sCJD are typically younger patients with a longer disease course [26], V carriers with FFI have a longer disease course, and gCJD patients with the V genotype have an earlier onset [27,28,29]. Among OPRIs patients, however, V carriers tend to experience a later onset and shorter disease duration [5,30]. Similar results were observed for previously reported 7-OPRI cases [7]. However, the specificity of this pedigree and the small number of cases prevented us from finding an association between codon 129 and disease duration in our pooled analysis; instead, only cis-V 129 showed a tendency for later onset.

To the best of our knowledge, modification in the amino-acid sequence of the octapeptides has not been reported in other patients with OPRIs. The identification of this novel insertional mutation that was segregated from the disease in our pedigree suggests that the mutation is likely pathological. The majority of affected individuals in our pedigree displayed the classic CJD phenotype, which indicated that the pathogenicity was not substantially altered by the single amino acid substitution. However, they also shared phenotypic features that may be specific to this genotype, notably the rapid disease progression. The cases in our pedigree (129 cis-M with AAS within OPRIs) had a significantly later onset and shorter disease duration compared to previously reported 7-OPRI cases with 129 cis-M without the AAS. Therefore, the conversion from glycine to glutamic acid in the last repeat of octapeptide insertion might provide a plausible hypothesis that the disease may progress more rapidly amongst patients with this genetic variant. Nonetheless, follow-up functional studies are needed to further elucidate this understudied phenomenon.

An interesting finding was that all cases of 7-OPRI with a short duration of disease had varying degrees of hyperintensity on brain MRI, which was not observed in patients with a longer duration of disease. This suggests that hyperintensity on brain MRI may indicate more rapid disease progression. In fact, there is evidence that hyperintensity on brain MRI of prion disease patients may be partially related to the degree of spongiosis and PrP^Sc^ deposition [31,32], which are associated with the severity of the disease. It is, therefore, reasonable to surmise that patients with a shorter disease duration have a faster propagation of prions and/or production of neurotoxic forms of PrP^Sc^ and are prone to hyperintensity on brain MRI. Consistent with our findings, in a study of 69 cases of gCJD with E200K, the degree of cortical hyperintensity was found to be negatively correlated with the duration of disease (r = −0.436, *p* = 0.002) [33].

Pathological changes in 7-OPRI carriers are also highly variable, with spongiosis occurring in the majority of cases and multicentric or elongated PrP plaque deposits in the cerebellum appearing in more than half of the cases. These pathological changes are associated with the disease phenotype. We found that 7-OPRI patients with spongiosis and multicentric PrP plaques had a later age of onset and shorter disease duration, which differs from the long course of GSS caused by point mutations. We hypothesize that this may be related to the fact that 7-OPRI patients usually have a combination of moderate to severe spongiosis, whereas GSS is associated with little or no spongiosis. The co-existence of these two pathologies may accelerate disease progression. The impact of the codon 129 polymorphism on disease pathology also cannot be ignored, as two of the three patients were 129 cis-V carriers.

There are a few limitations to this study that should be taken into account. First, despite the fact that all four patients included in the analysis had the 7-OPRI mutation, a pathological examination was conducted for only one patient. This, however, is an unfortunate and unavoidable reality in China due to traditional ethical values. I1 and II2 were not genetically confirmed, and it is possible that not all affected family members had the same 7-OPRI composition. Second, although reconstruction of the three-dimensional structure revealed that the AAS within OPRIs affected the structure of the protein, no in vitro functional tests were performed. Third, given the small number of cases, the findings of this study should be interpreted with caution and validated in other 7-OPRI families.

In conclusion, we have reported three cases of 7-OPRI with an AAS within OPRIs, which extends the genotype and clinical phenotype of 7-OPRI and confirms the heterogeneity of clinical presentation in the same family. The heterogeneity in the duration of the disease may depend on the codon 129 genotype and AAS within OPRIs.

## Figures and Tables

**Figure 1 viruses-14-02245-f001:**
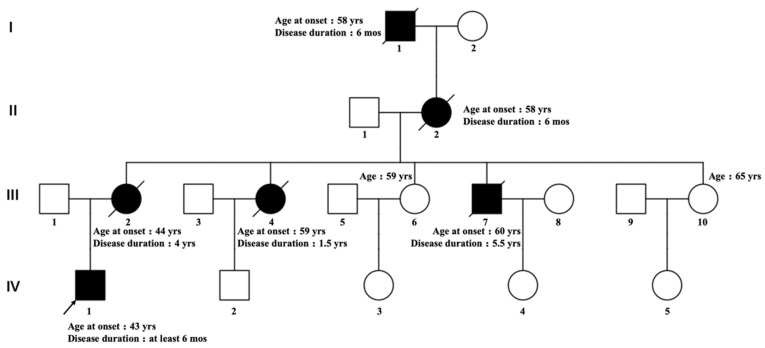
The pedigree of Chinese family members affected by 7-OPRI. The proband is indicated with an arrow. Black color indicates the individuals who presented symptoms of dementia. The generations are marked with numbers, with I being the first generation, II being the children of the first generation, and III being the grandchildren, and so on.

**Figure 2 viruses-14-02245-f002:**
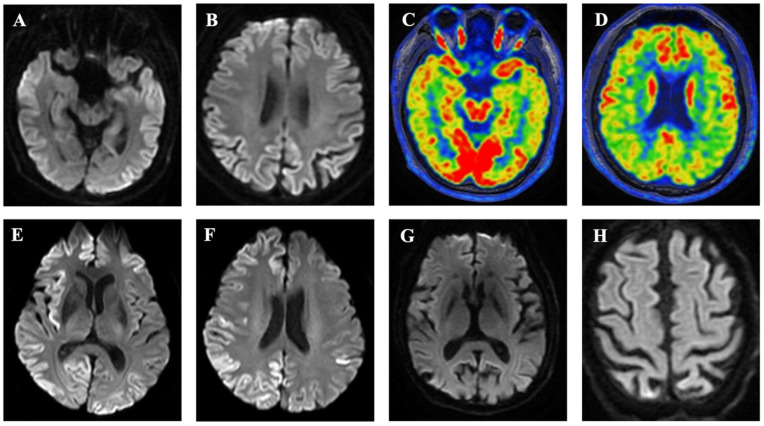
(**A**,**B**) MRI DWI data obtained on day 7 after the onset of symptoms in case IV-1 indicating bilateral symmetrical hyperintensity in the fronto-temporoparietal-occipital cortex and subcortex. (**C**,**D**) The ^18^F-FDG PET data also showed significant hypometabolism in the aforementioned areas. (**E**,**F**) MRI DWI data obtained on day 14 after the onset of symptoms in case III-4 indicating bilateral frontotemporal parietal occipital hyperintensity, with the right side being more prominent. (**G**,**H**) Brain MRI showed brain atrophy without hyperintensity in case III-7.

**Figure 3 viruses-14-02245-f003:**
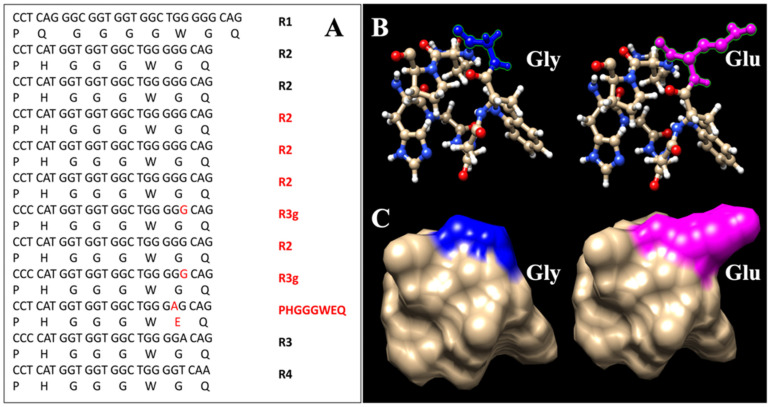
(**A**) Nucleotide sequence of the mutant allele in the region between codons 51 and 91 of the *PRNP* in our pedigree. The 7-OPRI are inserted after the second R2 cycle. The mutated nucleotide and amino acid are shown in red. (**B**) The PHGGGWGQ residues are shown as a cartoon model. The wild-type Gly residue is indicated in blue. The mutated Glu residue with acidic side chains in our patient is indicated in magenta. (**C**) These residues are shown as a surface model. The wild-type Gly residue, which has hydrophobic side chains, is indicated in blue. The mutated Glu residue found in our patient is indicated in magenta.

**Figure 4 viruses-14-02245-f004:**
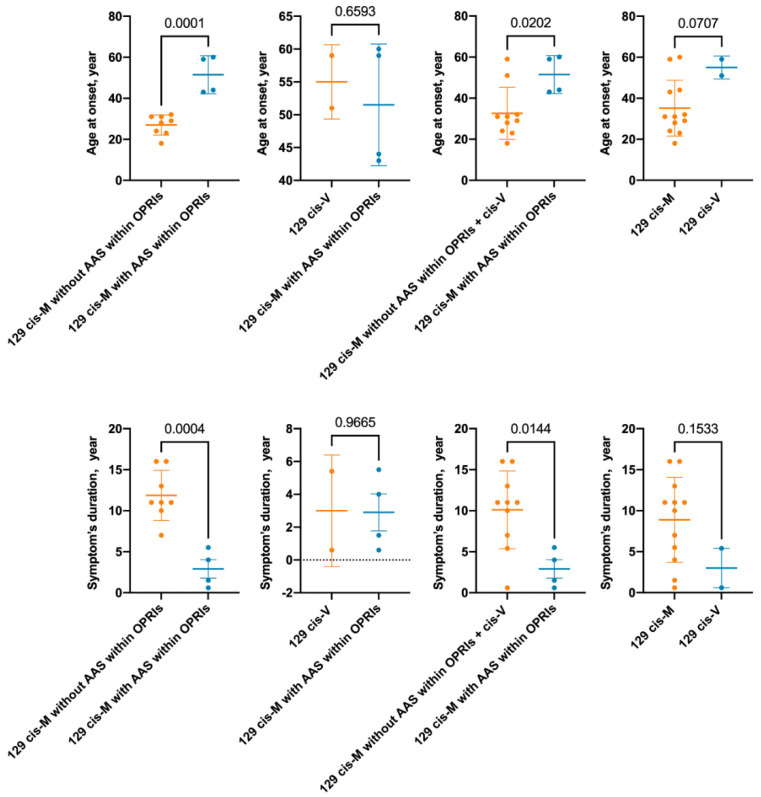
Comparison of age at onset and duration of symptoms in patient groups demarcated by codon 129 genotype and ASS within OPRIs. ASS, amino acid substitution; OPRIs: octapeptide repeat insertions.

**Table 1 viruses-14-02245-t001:** Comparison of clinical and auxiliary characteristics of diseases with short-term and long-term courses.

	Total Patients (*n* = 17)	Short Clinical Course (*n* = 4)	Long Clinical Course (*n* = 13)	*p*
**Baseline characteristics**				
Female, %	7/15 (46.7)	2/4 (50.0)	5/11 (45.5)	1.000
Age at onset, year, mean ± SD	40.6 ± 14.4	53.0 ± 7.3	35.0 ± 12.9	0.021
Symptom’s duration, year, mean ± SD	7.3 ± 5.2	0.9 ± 0.4	9.2 ± 4.3	<0.001
Family history, %	16/17 (94.1)	4/4 (100.0)	12/13 (92.3)	1.000
**Initial symptoms**				
Cognitive defects, %	11 (64.7)	3 (75.0)	8 (61.5)	1.000
Psychiatric disturbances, %	6 (35.3)	1 (25.0)	5 (38.5)	1.000
Ataxia, %	4 (25.3)	1 (25.0)	3 (23.1)	1.000
Speech disturbances, %	3 (17.6)	2 (50.0)	1 (7.7)	0.121
**Clinical features of prion diseases**				
Cognitive dysfunction, %	17 (100.0)	4 (100.0)	13 (100.0)	1.000
Psychiatric disturbances, %	13 (76.5)	3 (75.0)	10 (76.9)	1.000
Parkinsonism, %	12 (70.6)	3 (75.0)	9 (69.2)	1.000
Cerebellar signs, %	12 (70.6)	2 (50.0)	10 (76.9)	0.538
Myoclonus, %	8 (47.1)	3 (75.0)	5(38.5)	0.294
Pyramidal signs, %	8 (47.1)	2 (50.0)	6 (46.2)	1.000
Speech disorders, %	6 (35.3)	2 (50.0)	4 (30.8)	0.584
Mutism, %	4 (23.5)	2 (50.0)	2 (15.4)	0.219
Seizure, %	3 (17.6)	0	3 (23.1)	0.541
Visual signs, %	2 (11.8)	0	2 (15.4)	1.000
**Laboratory features**				
PSWCs on EEG, %	2/11 (18.2)	1/3 (33.3)	1/8 (12.5)	0.491
Positive CSF 14-3-3 protein, %	2/3 (66.7)	2/2 (100.0)	0/1	0.333
Elevated CSF tau protein, %	2/3 (66.7)	2/2 (100.0)	0/1	0.333
Positive RT-QuIC, %	2/2 (100.0)	2/2 (100.0)	0/0	-
Hyperintensity on MRI, %	3/7 (41.9)	3/3 (100.0)	0/4	0.029
Condon 129				0.066
129 cis-M	12/14 (85.7)	2/4 (50.0)	10/10 (100.0)	
129 cis-V	2/14 (14.3)	2/4 (50.0)	0/10	
AAS within OPRIs				0.516
Yes	4/15 (26.7)	2/4 (50.0)	2/11 (18.2)	
No	11/15 (73.3)	2/4 (50.0)	9/11 (81.8)	
Neuropathology				
Spongiosis	9/11 (81.8)	1/1 (100.0)	8/10 (80.0)	1.000
Multicentric plaques	3/11 (27.3)	1/1 (100.0)	2/10 (20.0)	0.273
Elongated plaques	3/11 (27.3)	0/1	3/10 (30.0)	1.000
Kuru-like plaques	1/11(9.1)	0/1	1/10 (10.0)	1.000
Nonspecific change	1/11 (9.1)	0/1	1/10 (10.0)	1.000

AAS: amino acid substitution; CSF: cerebrospinal fluid; EEG: electroencephalogram; MRI: magnetic resonance imaging; OPRIs: octapeptide repeat insertions; PSWCs: periodic sharp wave complexes; RT-QuIC: real-time quaking-induced conversion.

## Data Availability

Data are available on reasonable request. The data in this research can be obtained by email to chenzhongyun3@163.com.

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
