# Peer review of "Amino Acid Substitution within Seven-Octapeptide Repeat Insertions in the Prion Protein Gene Associated with Short-Term Course"

_viruses, 2022, doi:10.3390/v14102245_

Round 1

Reviewer 1 Report

The authors have given P values in Fig 4 sometimes on extremely low numbers.  I think the methodology should be checked by a statistical reviewer.

The last sentence on p8 incorrectly states  '129 cis-M cases tended to have a later age of onset '.  In Fig 4 the reverse is shown.

Author Response

The authors have given P values in Fig 4 sometimes on extremely low numbers.  I think the methodology should be checked by a statistical reviewer.

Reply: Thank you for your suggestion. We strongly agree with you. The tiny sample size is one of the limitations of this study. Therefore, among the limitations of this study, we emphasize that the results of this study should be interpreted with caution. Nevertheless, in the groups that differed statistically, the overall sample size was greater than 10. Therefore, we believe that this result is still somewhat instructive. As 7-OPRI patients are very rare, these results still need to be validated in other OPRIs patients. 

The last sentence on p8 incorrectly states  '129 cis-M cases tended to have a later age of onset '.  In Fig 4 the reverse is shown.

Reply:  Thank you for your suggestion. We have revised it accordingly.

Reviewer 2 Report

Chen et al. report clinical and paraclinical presentation of three Chinese family members carrying a novel 7-OPRI with an amino acid substitution. Chen et al. also review published literature concerning 7-OPRI cases worldwide and discuss disease phenotype and genotype influence on overall disease presentation, i.e. age at onset and disease duration. Based on literature review and own observations Chen et al. elegantly suggest that the novel mutation could be associated with short disease duration. 

I do not have recommendations for major changes or supplementary materials. Just a couple comments:

- Add genetic analysis results, i.e. c129 and 7-OPRI composition, for the patient III-7.

-  Discuss the possibility that not all affected family members had the same 7-OPRI composition.  

Author Response

Add genetic analysis results, i.e. c129 and 7-OPRI composition, for the patient III-7.

Reply: Thank you for your suggestion. We accidentally omitted to mention III-7's genotype.  In this study, genetic analysis was performed for patients III-2, III-4, III-7 and â…£-1 (page 4, first paragraph). All the patients had MM at locus 129 and carried the 7-OPRI mutation.

-  Discuss the possibility that not all affected family members had the same 7-OPRI composition.

Reply: Thank you for your suggestion. I1 and II2 were not genetically confirmed in our study. In the statistical analysis, these two cases were not included. We have added them to the “limitation” section of this study.